# Evaluation the Effect of Sonodynamic Therapy with 5-Aminolevulinic Acid and Sodium Fluorescein by Preclinical Animal Study

**DOI:** 10.3390/cancers16020253

**Published:** 2024-01-05

**Authors:** Chiung-Yin Huang, Jui-Chin Li, Ko-Ting Chen, Ya-Jui Lin, Li-Ying Feng, Hao-Li Liu, Kuo-Chen Wei

**Affiliations:** 1Neuroscience Research Center, Chang Gung Memorial Hospital, Taoyuan 33305, Taiwan; chyinhuang@cgmh.org.tw (C.-Y.H.); rock3345678@gmail.com (J.-C.L.); chenkoting@gmail.com (K.-T.C.); a9360@cgmh.org.tw (Y.-J.L.); lynn74tw@gmail.com (L.-Y.F.); 2Department of Neurosurgery, New Taipei Municipal TuCheng Hospital, New Taipei 236017, Taiwan; 3Department of Neurosurgery, Chang Gung Memorial Hospital, Taoyuan 33305, Taiwan; 4School of Medicine, Chang Gung University, Taoyuan 33302, Taiwan; 5Division of Natural Product, Graduate Institute of Biomedical Sciences, Chang Gung University, Taoyuan 33302, Taiwan; 6Department of Electrical Engineering, National Taiwan University, Taipei 10617, Taiwan

**Keywords:** sonodynamic therapy, brain tumor, focused ultrasound, 5-ALA, fluorescein

## Abstract

**Simple Summary:**

The use of noninvasive sonodynamic therapy (SDT) for brain tumor treatment is considered to be highly potential. The sonosensitizer is the key component of SDT; choosing an appropriate sonosensitizer will help to enhance the treatment response. In this study, somatic and brain tumor model animals were used to evaluate the effect of two sonosensitizers, 5-ALA and fluorescein. The data suggest that the tumor inhibitory effect of both sonosensitizers is equivalent in subcutaneous tumors. In early-stage brain tumors with relatively intact blood–brain barriers, 5-ALA penetrates well and promotes tumor inhibitory effect by SDT, while fluorescein fails to accumulate in tumor area and no therapeutic effect was observed. In conclusion, both fluorescein and 5-ALA are safe and effective SDT sonosensitizers, and the tumor microenvironment and pathologic type should be considered in the selection of adequate sonosensitizers.

**Abstract:**

Sonodynamic therapy (SDT) is a novel tumor treatment that combines biosafe sonosensitizers and noninvasive focused ultrasound to eradicate solid tumors. Sonosensitizers such as 5-aminolevulinic acid and fluorescein have great potential in tumor treatment. Here, rodent subcutaneous and brain tumor models were used to evaluate the treatment effect of both 5-ALA- and fluorescein-mediated SDT. The subcutaneous tumor growth rates of both SDT groups were significantly inhibited compared with that of the control groups. For intracranial tumors, 5-ALA-SDT treatment significantly inhibited brain tumor growth, while fluorescein-SDT exerted no therapeutic effect in animals. The distribution of fluorescein in the brain tumor region underwent further assessment. Seven days post tumor implantation, experimental animals received fluorescein and were sacrificed for brain specimen collection. Analysis of the dissected brains revealed no fluorescence signals, indicating an absence of fluorescein accumulation in the early-stage glioma tissue. These data suggest that the fluorescein-SDT treatment response is closely related to the amount of accumulated fluorescein. This study reports the equivalent effects of 5-ALA and fluorescein on the treatment of somatic tumors. For orthotopic brain tumor models, tumor vascular permeability should be considered when choosing fluorescein as a sonosensitizer. In conclusion, both fluorescein and 5-ALA are safe and effective SDT sonosensitizers, and the tumor microenvironment and pathologic type should be considered in the selection of adequate sonosensitizers.

## 1. Introduction

Malignant glioma is a common and severe primary brain tumor with an extremely poor prognosis. The median overall survival is approximately 1 year, and the survival rates at 5 and 10 years are very low (approximately 5% and 3%, respectively) [1]. Therefore, the development of new treatment strategies is urgently needed. Recently, sonodynamic therapy (SDT) has shown great potential for brain tumor treatment. SDT is an emerging therapeutic approach that utilizes low-intensity ultrasound and a sonosensitizer to selectively kill cancer cells or other pathological tissues [2,3]. By combining ultrasound waves and sonosensitizers, SDT induces the production of reactive oxygen species (ROS) and causes damage to proteins, nucleic acids, lipids, membranes and organelles, leading to tumor cell death [4]. This newly explored modality shows promise as a noninvasive treatment for brain tumors. SDT has shown promising results in preclinical studies, increasing tumor growth inhibition and improving survival rates in animal models [5,6,7,8]. The combination of ultrasound waves and sonosensitizers enables deeper penetration into tissues, targeting cancer cells that are difficult to reach by photodynamic therapy. By using intracranial focused ultrasound (FUS), it is possible to treat deep-seated brain tumors noninvasively. However, further research is needed to optimize the treatment parameters and evaluate the safety and efficacy in clinical trials.

The use of focused ultrasound (FUS) provides an opportunity to implement central nervous system (CNS) SDT clinically. FUS is a noninvasive, targeted ultrasonic wave treatment that produces thermal or mechanical effects on a specific area of tissue without damaging surrounding healthy tissues [9,10,11]. FUS has many advantages for brain disease treatment. First, FUS can noninvasively penetrate the skull and reach deep structures within the brain, enabling precise targeting of brain tumors or other abnormalities [12]. Second, FUS can be combined with microbubbles to open the blood–brain barrier (BBB) to increase drug delivery. Finally, the use of FUS is simple and can be repeated for patients who need repeated treatments. These advantages make FUS a promising tool for brain disease treatment, as well as for the potential applications in brain tumor SDT treatment [10].

Regarding the CNS, studies have demonstrated the efficacy of SDT in preclinical models of glioblastoma multiforme (GBM), the most common and aggressive type of brain tumor. SDT has shown improved tumor growth inhibition and prolonged survival rates in animal models of GBM [13,14,15]. One of the most essential elements for SDT are sonosensitizers, which can selectively accumulate in tumor cells and be activated by FUS to eliminate cancer cells. There are various types of sonosensitizers, including porphyrin-based sonosensitizers, xanthene-based sonosensitizers, nonsteroidal anti-inflammatory drug-based sonosensitizers, and other sonosensitizers [16,17]. In CNS SDT, two commonly used agents for fluorescence-guided neurosurgery, fluorescein and 5-aminolevulinic acid (5-ALA), are considered suitable sonosensitizers for SDT therapy. The 5-ALA agent is a porphyrin precursor in the heme synthesis pathway and is intracellularly metabolized to the fluorescence product protoporphyrin IX (PpIX) in the mitochondria. In cancer cells, the heme-synthesis enzyme porphobilinogen deaminase is upregulated to increase PpIX production. On the other hand, dysfunction of the ferrochelatase enzyme results in failure to produce heme from PpIX. Consequently, PpIX accumulates in cancer cells [5,18]. This tumor tissue-specific fluorescence enables guidance for tumor surgery or provides a sonosensitizing effect for SDT. Another potential sonosensitizer is the biosafe xanthene dye fluorescein. In contrast to 5-ALA, accumulation of fluorescein occurs through selective extravasation to the interstitial space under conditions of an abnormal tumoral vasculature structure [19,20]. Therefore, the use of fluorescein is not restricted by cancer cell metabolic conditions and uptake ability. Since the distribution and accumulation mechanisms of sonosensitizers are different, the choice of sonosensitizer may contribute to the efficacy of SDT. The 5-ALA-PpIX agent accumulates in tumor cells via metabolic processes within the target cells, while fluorescein nonspecifically penetrates because of tumor-induced BBB leakage and is distributed around tumor cells. These different mechanisms and characteristics may be applicable to the treatment of brain tumors with various pathological features. The purpose of this study was to compare the two sonosensitizers to determine their respective advantages and limitations for SDT in brain tumors. Such a study could help to optimize the choice of sonosensitizer for personalized SDT treatment of brain tumors.

## 2. Materials and Methods

### 2.1. Cell Culture

Rat C6 glioma cells (ATCC, Manassas, VA, USA) were used to establish ectopic and orthotopic brain tumor models. The glioma cells were cultured in Dulbecco’s modified Eagle’s medium (DMEM; Thermo Fisher, Waltham, MA, USA) supplemented with 100 U/mL penicillin, 100 μg/mL streptomycin, and 10% fetal bovine serum (Thermo Fisher, USA) at 37 °C with 5% humidified CO_2_.

### 2.2. Animal Preparation

All animal experiments were approved by the Institutional Animal Care and Use Committee of Chang Gung University and followed the experimental animal care guidelines (CGU110-032). Sprague–Dawley rats (6–8 weeks, 200–220 g) were used to establish animal tumor models. For the subcutaneous tumor model, 3 × 10^6^ C6 cells suspended in 100 µL of PBS were inoculated into the hind flank of each animal. Tumor volume was measured by digital calipers and calculated according to the formula V = ½ × (length × width^2^). The tumor growth rate was calculated based on the initial measurement; subsequent measurements were divided by the first measurement, then multiplied by 100% to determine the tumor growth rate at each time point. For the intracranial tumor model, animals were fixed with a stereotaxic frame, then a hole was created in the exposed cranium 1.5 mm anterior and 2 mm lateral to the bregma using a 27G needle. A total of 5 × 10^5^ cells suspended in 5 µL of PBS were infused into the brain striatum at a rate of 1 µL/min using a microdialysis pump system (CMA Microdialysis, Sollentuna, Sweden). After the infusion procedure, the needle was withdrawn over 2 min to prevent tumor cell reflux. A 9.4-Tesla magnetic resonance scanner (BioSpec 94/20 USR, Bruker, OK, USA) was used to monitor tumor growth every 7 days.

### 2.3. FUS Sonication

Animals were anesthetized with a mixture of oxygen (with a flow rate of 0.8 L/min) and 2% vaporized isoflurane using an anesthesia vaporizer. Each animal was placed directly under an acrylic water tank with its head attached tightly to a thin-film 4 × 4 cm^2^ window at the bottom of the tank. An FUS transducer (Sonic Concepts, Bothell, WA, USA, H-107) driven by an arbitrary function generator (33250A, Agilent, Santa Clara, CA, USA) with a radio-frequency power amplifier (240 L, E&I, Jericho, NY, USA) was used for radio-frequency (RF) signal amplification. The focused ultrasound parameters used in this study are as follows: frequency = 500 kHz, peak pressure = 0.25/0.3/0.35/0.4 MPa, PRF = 10 Hz, burst duration = 10 ms, duty cycle = 10%, and total exposure time = 20 min.

### 2.4. Tumor Fluorescein Accumulation Imaging

Three-week-old Sprague–Dawley rats were fed fluorescence-free chow (Oriental Yeast Co., Ltd., Tokyo, Japan) for 4 weeks, after which subcutaneous or brain tumors were induced as previously described. Throughout the entire experimental period, the animals continued to be fed fluorescence-free chow. At 10 days post tumor implantation, 16 mg/kg of body weight of fluorescein was intraperitoneally injected. The subcutaneous tumor and whole brain were dissected 20 min after fluorescein administration. Fluorescence imaging was then observed with an emission wavelength of 550 nm and an excitation wavelength of 405 nm using an IVIS instrument (Spectral Instrument Imaging, Tucson, AZ, USA, Lago X).

### 2.5. Immunohistological Examination

To confirm the cytotoxic effect of SDT, animals were sacrificed 48 h after treatment. Paraformaldehyde-fixed and paraffin-embedded tumors were sliced into 4 μm thick sections for IHC analysis. Sections were deparaffinized by being placed in hot xylene for 10 min, then rehydrated by descending grades of alcohol. To block endogenous peroxidase activity, tissue sections were incubated in 3% H_2_O_2_ for 10 min at room temperature. Antigen retrieval was performed via incubation in citrate buffer at 92 °C for 20 min. Cleaved caspase-3 (Cell Signaling, Danvers, MA, USA; #9664, 1:500), survivin (Cell Signaling; #2808, 1:400), and Ki67 (Abcam, Cambridge, UK, ab16667, 1:200) antibodies were employed to monitor tumor proliferation and apoptosis.

### 2.6. Statistical Analysis

The data are expressed as the mean ± SD on the basis of at least three independent experiments. Statistical analysis was performed using Student’s t test and log-rank test. Statistical significance was defined as *p* < 0.05. All statistical analyses were performed in GraphPad Prism 10.0.3. licensed to C.Y.H.

## 3. Results

### 3.1. SDT with Fluorescein in Rat Subcutaneous Tumors

The power levels of FUS for SDT sonication were examined in rat subcutaneous tumors. Seven days post tumor cell inoculation, the tumor size was measured, and the animals were divided into eight groups (control, fluorescein alone, FUS 0.25 MPa, FUS 0.3 MPa, 0.35 MPa, FUS 0.25 MPa with fluorescein, FUS 0.3 MPa with fluorescein, and FUS 0.35 MPa with fluorescein). To proceed with SDT, animals were intraperitoneally administered 16 mg/kg body weight fluorescein 20 min prior to FUS treatment. Three ultrasound exposure conditions with increasing acoustic intensities were applied, and tumor size was measured every two days. As shown in Figure 1, there were no significant differences in tumor volume among the fluorescein-only, FUS-only, and control groups. The tumor volumes in the fluorescein-SDT groups with all three FUS exposure conditions were significantly smaller than that in the control group (tumor growth ratio, control vs. 0.25 MPa + FL: 0.0088, control vs. 0.3 MPa + FL: 0.0057, control vs. 0.35 MPa + FL: 0.0012). Furthermore, a significant difference was observed between the fluorescein-only and FUS-only groups (Appendix A). These results indicate that treatment with ultrasound combined with fluorescein inhibited the growth of subcutaneous tumors in vivo (Figure 1). Furthermore, the inhibitory effect was positively correlated with FUS intensity in a dose-dependent manner. The body weight of all animals was also monitored during the experimental periods, and no significant decrease was recorded. These results also indicate the safety of fluorescein-SDT in animal models.

### 3.2. Histological Examination of the Antitumor Effect of Fluorescein-SDT

Histological staining was performed to analyze the tumor tissue for determination of the fluorescein-SDT antitumor effect. Tumor tissue samples were collected 2 days post SDT treatment, and the expression of survivin, Ki67, and active caspase-3 was examined by immunohistochemical staining. Survivin is expressed in most human malignancies and protects cells from apoptosis by inhibiting the apoptosis protein family. The expression of survivin was lower in the fluorescein-SDT groups than in the control and fluorescein-only groups, demonstrating the potential activation of the apoptosis pathway. The decreased expression of Ki67 also revealed inhibition of cell proliferation. Caspase-3 is the major effector caspase and is responsible for the cleavage and activation of other caspases in photodynamic therapy-associated apoptosis. The staining results also showed significant overexpression signals of activated caspase-3, which indicated that the apoptosis pathway was activated after fluorescein-SDT administration (Figure 2).

### 3.3. SDT with 5-ALA in Rat Subcutaneous Tumors

To study the efficacy of 5-ALA as a sonosensitizer for SDT, rat subcutaneous tumor model animals were treated orally with 5-ALA (180 mg/kg body weight) 4 h prior to the experiment. Then, animals received FUS treatment (0.4 MPa, duty10%, 20 min) for SDT. As shown in Figure 3, the tumors of the 5-ALA-SDT groups were significantly smaller than those of the control, 5-ALA-only, and FUS-only groups (control vs. 5-ALA-SDT: *p* = 0.0121, FUS-only vs. 5-ALA-SDT: *p* = 0.0011, 5-ALA-only vs. 5-ALA-SDT: 0.0022). These results indicate that when experimental animals are treated with FUS alone or 5-ALA alone, there is no therapeutic effect on the tumor. Therapeutic efficacy can only be achieved when FUS and 5-ALA administration are combined. Furthermore, the effect of 5-ALA and fluorescein as sonosensitizers for SDT treatment is equivalent in treating subcutaneous tumor model animals.

### 3.4. SDT with 5-ALA in a Rat Brain Tumor Model

The glioma rats were randomly divided into control, 5-ALA-only, and 5-ALA-SDT groups for SDT treatment. On day 7, the animals were treated orally with 5-ALA (180 mg/kg body weight) 4 h prior to SDT (0.4 MPa, duty10%, 20 min) treatment. Figure 4A shows that the tumor growth rate of the 5-ALA-SDT group was significantly slower than those of the control, FUS-only, and 5-ALA-only groups (all groups *p* < 0.001 on day 20), and there were no significant differences among the control, FUS-only, and 5-ALA-only groups. The data in Figure 4B show that animals treated with 5-ALA-SDT had improved survival (*p* = 0.0025 vs. control). Figure 4C illustrates the T1-weighted images of the SDT-treated animals. The brain images of the animals in the control and 5-ALA-only groups on day 20 post tumor inoculation demonstrate that enlarged brain tumors occupied the right hemisphere, while the tumor mass in 5-ALA-SDT-treated animals was significantly reduced.

### 3.5. SDT with Fluorescein in a Rat Brain Tumor Model

To evaluate the efficacy of fluorescein-SDT on brain tumors, the glioma rats were randomly divided into control, FUS-only, fluorescein-only, and fluorescein-SDT groups. On day 7, the animals received 16 mg/kg body weight fluorescein by intraperitoneal injection. Twenty minutes later, SDT treatment was performed with the following parameters: 0.4 MPa, 10% duty time, and 20 min. As shown in Figure 5A, the tumor growth rate and animal survival proportions did not differ among all four groups (Figure 5B). Brain magnetic resonance (MR) images were obtained from each treatment subgroup to measure the effect of fluorescein-SDT on brain tumor volume (Figure 5C). To check the distribution of fluorescein in rat subcutaneous tumor and brain tumor, a different set of tumor-bearing animals was used to detect the fluorescein signals. To avoid interference of background fluorescence values on observation, animals were fed fluorescence-free chow for 3 weeks prior to tumor implantation. Seven days post tumor induction, both subcutaneous and brain tumor-bearing animals were intraperitoneally injected with 16 mg/kg body weight fluorescein, while controls were injected with saline. As shown in Figure 5D, fluorescence intensity images of subcutaneous tumor tissues dissected 20 min after injection showed strong accumulation, while no fluorescence signals were detected in control animals. In tumor-bearing rat brains, no signals were detected in either the control or fluorescein-administered groups. The results imply that the blockade of the BBB may prevent the distribution of fluorescein to the brain and brain tumor tissues in this stage of brain tumor.

## 4. Discussion

Sonosensitizers are key components of SDT, and their successful outcome depends on appropriate pharmacokinetic characteristics and the ability to specifically accumulate in the tumor. In this study, two potential sonosensitizers, fluorescein and 5-ALA, were used to treat models of two typical tumors (a subcutaneous tumor model and an orthotopic brain tumor model) to investigate the factors influencing SDT. Our experiment shows that both 5-ALA and fluorescein are effective in treating subcutaneous tumor models when used as sonosensitizers. However, in the treatment of brain tumors, 5-ALA-SDT remains effective while fluorescein-SDT fails. Further study of fluorescein distribution reveals insufficient permeability of early-stage brain tumor vasculature for fluorescein to accumulate in the tumor area. These data suggest that both sonosensitizers are effective for FUS, and the amount of sonosensitizer accumulated in the tumor region directly affects the outcome of SDT treatment.

Fluorescein has been used as a fluorescent guide for neurosurgery, and its selective extravasation to tumor areas make it suitable for use as a sonosensitizer. In the subcutaneous tumor model, three FUS intensity parameters were assessed for fluorescein-SDT. A significant tumor inhibition effect was shown in the fluorescein-SDT groups, and no significant differences were found between the control and FUS-only groups. These results clearly demonstrate that fluorescein is an effective sonosensitizer for SDT treatment and works well with FUS energies ranging from 0.25 MPa to 0.35 MPa. Although there was a trend of FUS energy dose-dependent effects in the fluorescein-SDT groups, no significant difference was observed among these three energy levels. The potential underlying mechanism of fluorescein-SDT was also studied by performing immunohistochemical staining of apoptosis-related markers and the cell proliferation marker Ki67 2 days post SDT treatment. In the fluorescein-SDT groups, lower expression of survivin revealed the progression of the apoptosis pathway after SDT treatment, and upregulation of signals of cleaved caspase-3 also indicated activation of the apoptosis pathway. The data suggest that fluorescein-SDT treatment induces apoptosis and suppresses tumor proliferation and invasion. These results are consistent with those of previous studies using 5-ALA and sinoporphyrin sodium as sonosensitizers [8,13,14,15], suggesting that fluorescein-SDT exerts an equivalent effect for tumor treatment. Here, using a subcutaneous tumor model, the reproducibility and reliability of fluorescein-SDT for tumor treatment were demonstrated, highlighting its potential for clinical use.

Both 5-ALA and fluorescein are used as fluorescent guiding agents for surgery. The accumulation mechanism of 5-ALA-PpIX is different from that of fluorescein, which relies on the altered metabolic pathway in cancer cells [21]. In the clinic, 5-ALA is used as an oral solution, which was approved by the European Medicines Agency (EMA) in 2007 [22]. A decade later, in 2017, the US Food and Drug Ad-ministration (FDA) also approved the use of 5-ALA for glioma surgery [20,23]. It is metabolized in the heme biosynthesis pathway to PpIX, then accumulates intracellularly in tumor cells. Previous multicenter studies have demonstrated the safety and effectiveness of 5-ALA, with minimal adverse side effects [24]. Here, we used both subcutaneous and orthotopic glioma models to examine the efficacy of 5-ALA-SDT for glioma treatment. The tumor growth of the 5-ALA-SDT groups was significantly suppressed. These results suggest that 5-ALA-SDT is a noninvasive and feasible treatment strategy for patients with malignant glioma. The safety and convenience of 5-ALA administration and the noninvasiveness of the FUS procedure make repeated 5-ALA-SDT treatment possible, revealing the great potential of this strategy in clinical use.

Regarding the selection of sonosensitizers, there are some concerns about the ability of the effector to accumulate. Previous in vitro and in vivo studies on 5-ALA-SDT have proven its efficacy for tumor treatment; however, the effect may be dependent on cell characteristics [8,25]. In our preliminary data, the accumulation ability of the fluorescent mid-product PpIX differed by cell type (Appendix A). Although most cancer cells reveal increased accumulation of PpIX, some cancer cells retain low detectable fluorescence signals [26]. Consequently, 5-ALA-SDT may fail in the treatment of these specific types of gliomas. To treat these types of tumor cells that lack 5-ALA accumulation ability, using fluorescein as a sonosensitizer will be an ideal strategy. Unlike 5-ALA, fluorescein is not able to penetrate the BBB [27]. The extravasation of fluorescein depends on the defective vascular architecture of the tumor microenvironment. In high-grade gliomas, a destroyed BBB enables the penetration of fluorescein and a high density of dilated and poorly differentiated blood vessels in the tumor site, together with a chaotic architecture and aberrant branching, causing uneven pressure levels in the tumor interstitial tissue. Therefore, fluorescein may concentrate specifically at the tumor site.

For low-grade glioma patients in whom fluorescein is used for neurosurgery guidance, low fluorescence signals have reported in diffuse tumor areas with low-density cellular infiltration [28,29]. Histological examination of the obtained surgical specimens has also shown that the deep staining represents high-grade glioma features such as prominent endothelial proliferation and dense tumor cells, while faint staining represents relatively sparse tumor cells and scattered endothelial proliferation [30]. These results indicate that the distribution of fluorescein fails to achieve the effective concentration in low-grade gliomas with a relatively intact BBB structure. Similar results have also been observed in animal brain tumor models. Do and his colleague reported that the BBB is disrupted at different tumor progression stages using a brain tumor model established by stereotactic tumor cell injection [31]. In the early stage of tumor initiation, the intact BBB blocked the penetration of Evans blue dye (EB) at 7 days post tumor inoculation. As the tumors grew, the EB signals were increased, becoming detectable, and this change corresponded with the disruption of the BBB. In our study, glioma cells were also inoculated into the rat brain using stereotactic intracerebral injection. The absence of detectable fluorescence signals at 7 days post tumor induction indicated an intact BBB may block the penetration of fluorescein. The Evans blue staining study conducted on mice brains with tumors, 9 and 29 days post tumor implantation, shows that the blood–brain barrier (BBB) is intact at the early stage and disrupted at the late stage (Appendix A). These results suggests that the tumor growth time window at 7 days may have been too short for the formation of a typical nonuniform and highly permeable blood–tumor barrier. Consequently, no significant tumor-inhibitory effect was observed in fluorescein-SDT-treated early-stage orthotopic brain tumor animals, and further tissue images also showed no detectable fluorescence signals in brain tumor sites. In contrast, an effective SDT treatment response was correlated with strong fluorescence signals in subcutaneous tumors. The data presented here clearly demonstrate that the accumulation of fluorescein in the tumor site is one of the key elements for successful SDT treatment, suggesting that tumor vascular permeability should be considered before choosing fluorescein as a sonosensitizer. It has been demonstrated that FUS can open BBB and enhance permeability of large molecules such as gadolinium-based contrast agents [32]. Schebesch et al. also reported that the fluorescein signal is correlated with contrast-enhanced T1-weighted MR sequences [27]. Thus, FUS-induced BBB opening may induce the similar distribution of gadolinium-based contrast agents and fluorescein. Therefore, opening the BBB by FUS prior to SDT treatment may help to enhance fluorescein accumulation in the low-grade tumor area, and may help to overcoming the low accumulation ability of fluorescein in the area of the intact BBB.

## 5. Conclusions

The principle of SDT is to combine a sonosensitizer and FUS to generate cytotoxic effects against tumor cells; therefore, the selection of a proper sonosensitizer is one of the major concerns for effective therapy. The distinct accumulation mechanisms of 5-ALA-PpIX and fluorescein may complement each other, offering alternative choices for clinicians treating patients with SDT. The selective accumulation of 5-ALA-PpIX in tumor cells reveals its great potential as a sonosensitizer of choice. For some specific tumor types that fail to accumulate 5-ALA-PpIX, fluorescein—which can accumulate in areas where the BBB is disrupted—may be an alternative choice of sonosensitizer. In this study, the effects of SDT with sonosensitizers exhibiting different tumor site accumulation mechanisms were evaluated. Given the promising tumor-inhibitory effects of SDT mediated by both sonosensitizers, there is high potential for their application in clinical therapy.

## Figures and Tables

**Figure 1 cancers-16-00253-f001:**
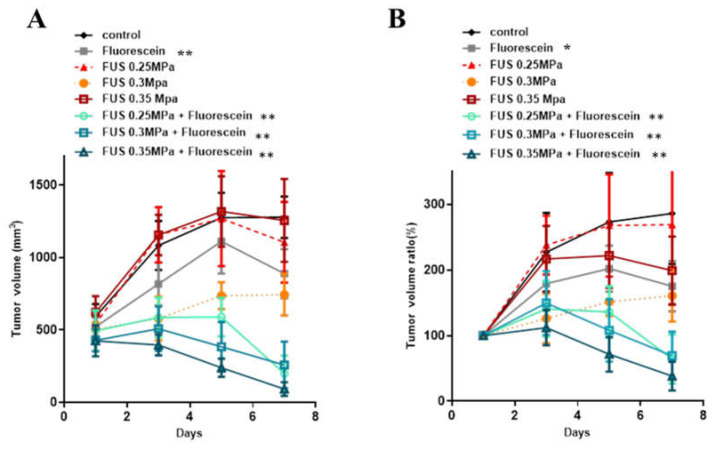
In vivo effects of SDT with fluorescein in subcutaneous C6 glioma tumors. The tumor volume (**A**) and normalized tumor growth ratio (**B**) were measured every 2 days after fluorescein-SDT treatment. The data are presented as the mean ± SD. * represents *p* < 0.05 verses control, and ** represents *p* < 0.01.

**Figure 2 cancers-16-00253-f002:**
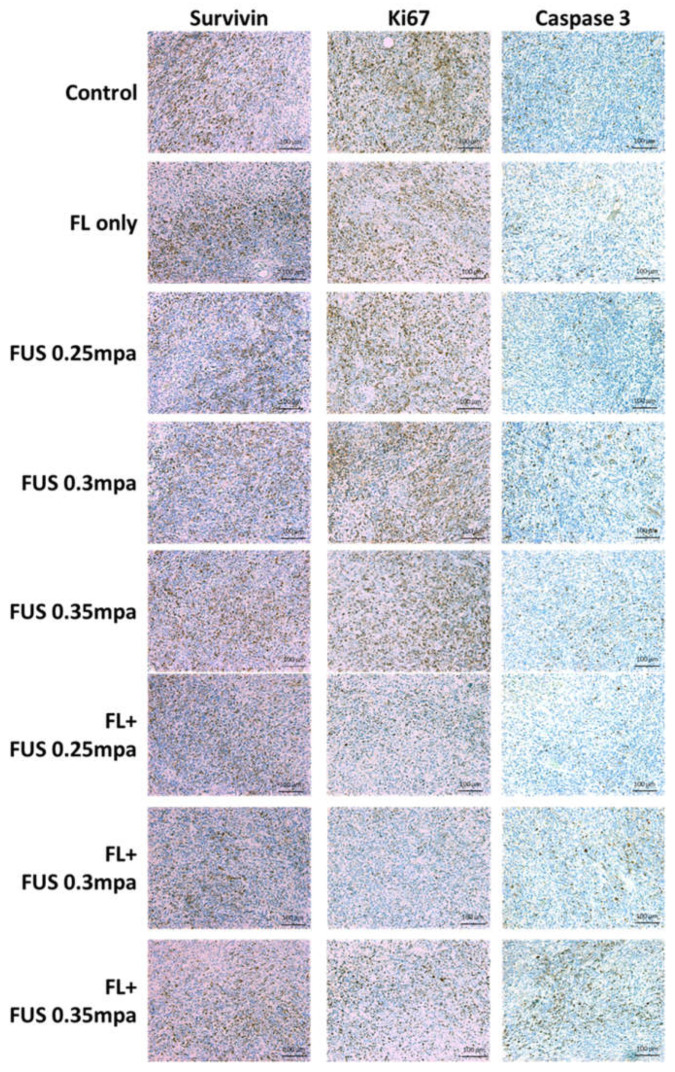
Immunohistochemical staining of survivin, Ki67, and activated caspase-3 in subcutaneous C6 tumors treated with fluorescein-SDT. The animals were sacrificed 48 h after treatment.

**Figure 3 cancers-16-00253-f003:**
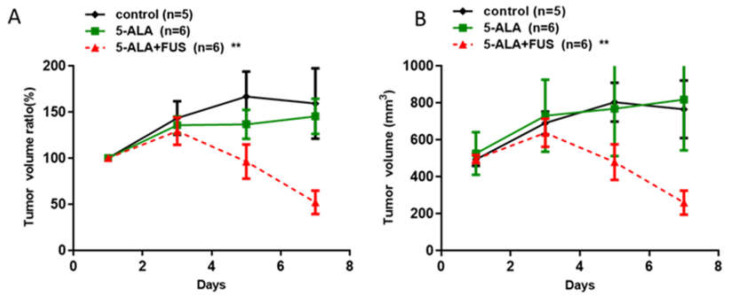
In vivo effects of SDT with 5-ALA in subcutaneous C6 glioma tumors. The tumor volume (**A**) and normalized tumor growth ratio (**B**) were measured every 2 days after 5-ALA-SDT treatment. The data are presented as the mean ± SD. ** represents *p* < 0.05 verses control.

**Figure 4 cancers-16-00253-f004:**
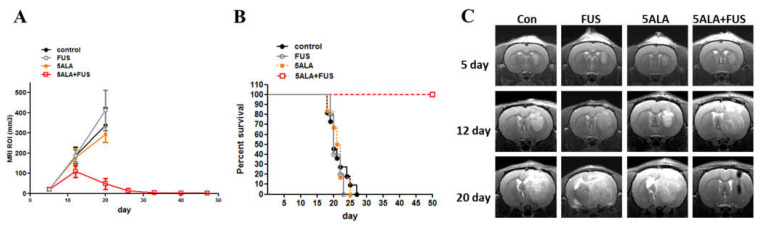
In vivo effects of SDT with 5-ALA in C6 brain tumors. (**A**) Treatment effect of 5-ALA-SDT. Tumor volume was measured in the region of interest (ROI) from 9.4T MRI imaging profiles. The data are presented as the mean ± SD. (**B**) Kaplan–Meier survival curves of brain tumor model animals with different treatments. (**C**) Representative T1-weighted images showing tumor growth in the different groups on days 5, 12, and 20 after tumor implantation. Tumors were implanted at day 0 and SDT treatment was performed on day 7; the arrow indicates SDT treatment day.

**Figure 5 cancers-16-00253-f005:**
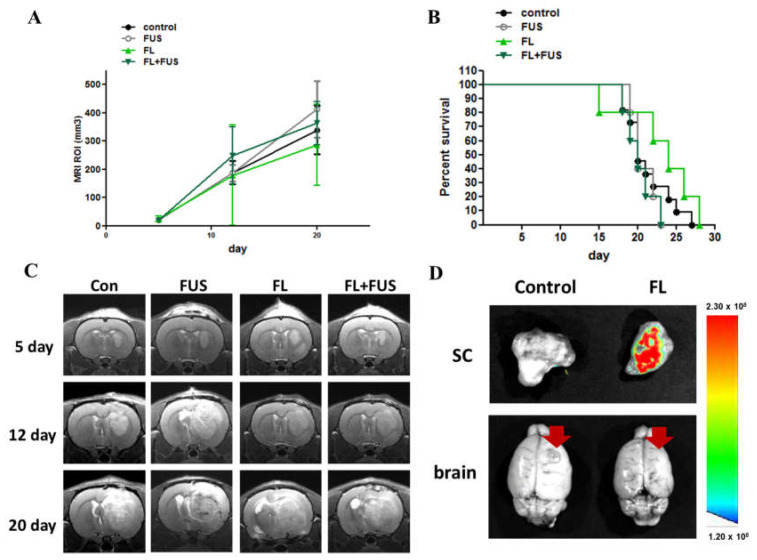
In vivo effects of SDT with fluorescein in C6 brain tumors. Fluorescein-SDT treatment was performed on day 7 post tumor implantation. (**A**) Treatment effect of fluorescein-SDT on tumor growth inhibition. The tumor volume was measured in the region of interest (ROI) from 9.4T MRI imaging profiles; the arrow indicates SDT treatment day. (**B**) Kaplan–Meier survival curves of brain tumor model animals with different treatments. (**C**) The representative T1-weighted images show tumor growth in the different groups on days 5, 12, and 20 after tumor implantation. (**D**) Photoimaging of subcutaneous tumors and brain tumors harvested 20 min post fluorescein administration. The red arrows indicate the tumor site and the color bar indicates the relative fluorescence.

## Data Availability

The raw data supporting the conclusions of this article will be made available by the authors on request.

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
