# Peer review of "Evaluation the Effect of Sonodynamic Therapy with 5-Aminolevulinic Acid and Sodium Fluorescein by Preclinical Animal Study"

_cancers, 2024, doi:10.3390/cancers16020253_

Round 1
Reviewer 1 Report
Comments and Suggestions for Authors
Dear Authors
Your manuscript describing the sonodynamic effects on different sonosensitizers in a preclinical study sounds interesting and suitable for the Journal. Overall, it looks that the experiments are performed well but the manuscript is written very superficial as a lot of info are missing or confusing. „We“ and „our“ should be reduced to an absolut minimum.
Questions and comments:
5-ALA as sonsosensitizer needs to be clarified as you mean 5-ALA-induced-PPIX as a sensitzer. This fact should be clearly stated throughout the manuscript. As consequence 5-ALA-induced-PPIX-SDT would be the correct term for the treatment. This should also be mentioned in the title.
L32: inhibited “brain” tumor growth
L33-34: It must be clarified that fluorescein showed not a cellular targeting accumulation or selectivity as it is a vessel related fluorophore. Please add info about the time of fluorescein application prior to the test.
L35: This statement is confusing as SDT is based on the concentration of the accumulated sonosensitizer and the statement is also related to 5-ALA-induced-PPIX.
L59: What do you mean with “deeper penetration”? Deeper compared to what?
L60: What are traditional therapies? Is PDT for GBM traditional? It looks not, there are some clinical papers available from French and German groups.
L66: “without damaging surrounding tissue”: That is not true as in case the FUS is focussed either on hard or on healthy tissue than there are damaging effects. Consequently, it is not selective and depends on the specialised physician performing FUS.
L67: What do you mean with advantages? Compared to what?
L72: “FUS is a noninvasive treatment method” this statement addresses the same info as the first advantage, thus it remains only two advantages?
L91: The tumor selectivity of 5-ALA-induced-PPIX is used for fluorescence guided resection which is approved by FDA and EU. Furthermore, it is now undergoing clinical investigation for Photodynamic Therapy in different concepts e.g. see French and German groups.
L121: Normally tumors have ellipsoidal or sphere shaped volume. Why did you use cuboid formula?
L134: What are the FUS-parameters used?
L142: Do you mean mg/kg body weight? If yes, mention it throughout the manuscript. If not, is it a concentration or xyz?
L143: “tumor and brain” Do you mean subcutaneous tumor and brain tumor?
L144: Specify the fluorescence excitation wavelength and the detection spectral range. IVIS is Perkin Elmer and Lago X is Spectral Instrument Imaging -> which one is used and what are the optical set ups and parameters for this specific experiment?
L175: control “shame OP” is missing
Figure 1: Please check the unit MPa throughout the manuscript and figures and revise. There is mixture of mpa – Mpa – MPa. Furthermore it looks that fluorescein alone (grey) has an impact of tumor growth by its own. Please mentioned and discuss. Furthermore FUS alone at FUS 0.3MPa may have a huge impact on the tumor growth. Why is that effect different to FUS 0.35MPa? What is the x-axe in day meaning? What happened on day 0? Please clarify whether on day 1 SDT was performed. How do you explain tumor growth until 2 days post treatment? How many individual animals were in each group? ** and * are not shown in the figure. Is there any inflammation due to FUS?
L208: 180mg/kg bw is not a clinical related 5-ALA dosage. Please comment on that and discuss possible side effects.
L210-212: This statement is only valid at day 7 on that day axe, where I dont know what happened, on day3 they showed similar volume - then there is a decrease. Is there any interpretation about the delayed tissue effect or tumor response?
L214: “administered together” please clarify as you mentioned 5-ALA is given 4h before FUS in the hope that a high concentration of PPIX is produced in the tumor cells.
L215-216: no they differ as with fluorescein is no increase on day 3!
L224-228: I think that these info should be transferred to the M&M section. In addition, the question arose what is a control in your term? There is need for several controls in these kind of experiments: shame operation, control without any treatment, control only SD, control only 5-ALA, plus the experimental group 5-ALA SDT
L229: Please define “growth rate“. In figure 4 simply the measured size of the tumors based on MRI-measurment is shown on some days after xyz-days (not defined as mentioned above).
Figure 4: Graph-A: X-axe: what is day? What is the treatment day? It sounds that d0 is xy, d5 ... please correlate the time with the x-axe of figure 4 and with the text L236. L238: I thought there is no fluorescein? Please crosscheck. Treatment day should be marked in Graph-A. Graph-C: MRI of day 12: 5-ALA+FUS looks pretty similar size to con. Why? What is the black-part on d20 5-ALA+FUS?
L246-251: Should be transferred to M&M-section
L251: “growth rate” see a comment L229.
L254-256: This part is more a discussion than a result. Please transfer.
L261: Why did you use 10mg/kg bw fluorescein instead of 16mg/kg bw like before? Is there any concentration related effect?
L266-267: or is there a complete different pharmacokinetic?
Figure 5A: How did you calculate the one death in FL-group prior to d20? When did you perform MRI related to the treatment? Legend: “Fluorescein-SDT treatment was performed on day 7 post-tumor implantation”. This sentence is confusing to the x-axe. Please clarify.
L283: What is C6/SD?
L284: What are the factors influencing SDT?
L285: “…. fluorescein … accumulation in tumor …” There is no real accumulation when using fluorescein -> extravasation due to leackage of vessels. Please clarify.
L292: Definition of „effective“ is missing.
L295: What are the sonodynamic energies? They are not mentioned sofar.
L295-297: The beginning of the sentence is said already 3 lines above, thus redundancy, and the combination with „and Prada et al … „ does not make sense. Please clarify and revise.
L314: and ealier in EU
L320: What do you mean with „eliminated“?
L329-330 and figure S2: This fact is shown in a lot of publications. Thus it is not new.
L333-334: This is speculation and needs experiments. Thus rephrasing is needed.
L334-335: Please add ref that fluorescein is not able to penetrate the BBB.
L335-336: what is the difference between defective vessels and BBB-leackage?
L336-340: needs references.
L363-364: the selective accumulation of each sonosensitzer is a key-element in this kind of treatment, not only for fluorescein.
L366-368: references are needed
L377-380: can be remove as this is not a conclusion
Supplement:
Table S1: Please add info that these are the p-values of the performed t-tests. Please explain ** and *.Furthermore there no need to duplicate the statistical results by mirroring along the diagonal.
Figure S1: The rational about body weight changes is missing.
Figure S2: y-axe unit should be nm instead of mm. What was the conc. of 5-ALA for this experiment? The header of the graph should be cancelled.
Reviewer 2 Report
Comments and Suggestions for Authors
This topic is interesting, but some points need revision:
- in the introductory section write better what the mechanisms of non-invasive sonodynamic therapy are.
- in the introduction section describe better what is the purpose of this paper.
- "Here, 317 we used both subcutaneous and orthotopic glioma models to examine the efficacy of 5-ALA-SDT for glioma treatment. The tumor growth of the SDT groups was significantly suppressed; furthermore, we observed that the tumor mass was eliminated in several rat brain tumors." There sentences seem misleading. Please revise them.
- "Our data clearly demonstrate that the accumulation of fluorescein in the tumor site is one of the key elements for successful SDT treatment, suggesting that tumor vascular permeability should be considered before choosing fluorescein as a sonosensitizer." What do the author suggest? angiography ?
Comments on the Quality of English LanguageMinor editing of English language required
Author Response
Reviewer 2:
This topic is interesting, but some points need revision:
- in the introductory section write better what the mechanisms of non-invasive sonodynamic therapy are.
Answer:
Thanks for comment, we have strengthened the part for mechanisms of non-invasive sonodynamic therapy accordingly.
- in the introduction section describe better what is the purpose of this paper.
- "Here, 317 we used both subcutaneous and orthotopic glioma models to examine the efficacy of 5-ALA-SDT for glioma treatment. The tumor growth of the SDT groups was significantly suppressed; furthermore, we observed that the tumor mass was eliminated in several rat brain tumors." There sentences seem misleading. Please revise them.
Answer:
Thanks for comment, we have rewritten this part to avoid misleading.
- "Our data clearly demonstrate that the accumulation of fluorescein in the tumor site is one of the key elements for successful SDT treatment, suggesting that tumor vascular permeability should be considered before choosing fluorescein as a sonosensitizer." What do the author suggest? angiography ?
Answer:
In brain images of patients with brain tumors, the fluorescein signals are similar to the results of contrast-enhanced T1-weighted MR imaging (Schebesch et al., Turk Neurosurg 26(2): 185-194, 2016. DOI: 10.5137/1019-5149.JTN.16952-16.0). Therefore, the authors suggest using MRI contrast imaging for evaluation. For the patients with enhanced T1-weighted MR signals, fluorescein can be used as sonosensitizer.
